# The Performance of GALAD Score for Diagnosing Hepatocellular Carcinoma in Patients with Chronic Liver Diseases: A Systematic Review and Meta-Analysis

**DOI:** 10.3390/jcm12030949

**Published:** 2023-01-26

**Authors:** Ming-Cheng Guan, Shi-Yu Zhang, Qian Ding, Na Li, Ting-Ting Fu, Gui-Xia Zhang, Qian-Qian He, Feng Shen, Tian Yang, Hong Zhu

**Affiliations:** 1Department of Medical Oncology, The First Affiliated Hospital of Soochow University, Suzhou 215006, China; 2Department of Hepatobiliary Surgery, Eastern Hepatobiliary Surgery Hospital, Second Military Medical University (Navy Medical University), Shanghai 200438, China; 3Eastern Hepatobiliary Clinical Research Institute, Third Affiliated Hospital of Navy Medical University, Shanghai 200433, China; 4School of Clinical Medicine, Hangzhou Medical College, Hangzhou 310014, China

**Keywords:** hepatocellular carcinoma, GALAD score, chronic liver disease, diagnosis, early detection

## Abstract

Background GALAD score, comprising five clinical parameters, is a predictive model developed for hepatocellular carcinoma (HCC) detection. Since its emergence, its diagnostic ability has been validated in different populations with a wide variation. Therefore, we conducted a meta-analysis to investigate its overall diagnostic performance in differentiating HCC in chronic liver diseases. Methods Eligible studies were searched in the *Web of Science*, *PubMed*, *Scopus*, *Ovid*, *Cochrane Library*, and *Embase* databases by 29 May 2022. Pooled sensitivity, pooled specificity, and area under the receiver operating characteristic curve (AUC) with the corresponding 95% confidence intervals (CI) were estimated. Results Fifteen original studies (comprising 19,021 patients) were included. For detecting any-stage HCC, GALAD score yielded an excellent ability, with pooled sensitivity, specificity, and AUC of 0.82 (95%CI: 0.78–0.85), 0.89 (95%CI: 0.85–0.91), and 0.92 (95%CI: 0.89–0.94), respectively. Notably, further analyses demonstrated a good diagnostic accuracy of GALAD score for identifying Barcelona Clinic Liver Cancer staging (BCLC) 0/A HCC, with a moderate sensitivity (0.73 (95%CI: 0.66–0.79)) and a high specificity (0.87 (95%CI: 0.81–0.91)); by contrast, only 38% of early-stage patients can be identified by alpha-fetoprotein, with an AUC value of 0.70 (95%CI: 0.66–0.74). Following subgroup analyses based on different HCC etiologies, higher sensitivities and AUC values were observed in subgroups with hepatitis C or non-viral liver diseases. For detecting BCLC 0/A HCC in the cirrhotic population, GALAD score had a pooled sensitivity, specificity, and AUC of 0.78 (95%CI: 0.66–0.87), 0.80 (95%CI: 0.72–0.87), and 0.86 (95%CI: 0.83–0.89). Conclusions We highlighted the superior diagnostic accuracy of GALAD score for detecting any-stage HCC with a high sensitivity and specificity, especially for early-stage HCC, with a relatively stable diagnostic performance. The addition of GALAD score into ultrasound surveillance may identify more HCC patients. Our findings imply the robust power of the GALAD score as a HCC screening or diagnostic tool, and it should be further validated by more studies with high quality.

## 1. Introduction

Hepatocellular carcinoma (HCC) is a common yet lethal malignancy, as is attested by 2020 global cancer statistics [1]. Usually, this kind of tumor occurs in the setting of chronic liver diseases (CLD), predominantly including hepatitis B virus (HBV)-infected, hepatitis C virus (HCV)-infected, and cirrhotic liver caused by various etiologies [2]. Simultaneously, concerns are also increased globally by the high prevalence of HCC in elderly men [2]. As an aggressive cancer prone to metastasis, HCC is generally not diagnosed until it has progressed to an intermediate or advanced stage with poor survival [3]. Reportedly, diagnosis at its early stage, when the tumor is resectable, could improve long-term survival, with a 5-year overall survival rate of >70% [2,4]. As such, regular screening and monitoring are recommended for individuals at high risk for HCC by international guidelines, including patients with cirrhosis or CLD [5,6,7].

Serological biomarker testing, with non-invasive, objective, and repeatable characteristics, is often used as a robust tool for assisting in HCC screening, monitoring, and early detection [8]. Of them, alpha-fetoprotein (AFP) serves as a tumor biomarker for clinical diagnosis of HCC. The diagnostic efficacy of AFP, however, remains controversial, with sensitivities ranging from 25% to 90% and specificities from 35% to 100% at the threshold value of 20 ng/mL [9,10]. The instability of AFP limits its utilization as a diagnostic tool for HCC. Moreover, lens culinaris agglutinin-reactive fraction of AFP (AFP-L3) and des-gamma-carboxy prothrombin (DCP; also called protein induced by vitamin K absence or antagonist-II (PIVKA-II)) are another two HCC-specific markers commonly used in clinical practice [11,12]. Generally, the three tumor markers have been utilized in various clinical settings in East Asian and Western countries [13,14]. Nonetheless, a single biomarker is not particularly effective at detecting HCC in its early stage.

Given the above opinions, Johnson PJ developed a novel diagnostic model based on gender, age, and the three above-mentioned biomarkers (named GALAD score) in a British cohort in 2014 [15]. The score was calculated as −10.08 + 1.67 × (Gender [1 = male, 0 = female]) + 0.09 × (Age) + 0.04 × (AFP-L3%) + 2.34 × log_10_(AFP) + 1.33 × log_10_(DCP). Since the first emergence of GALAD score, its diagnostic ability for detecting HCC has been verified in different populations. As such, we conducted this meta-analysis to comprehensively access the overall diagnostic performance of the GALAD score in differentiating any-stage or early-stage HCC in CLD patients. Subgroup analyses were also conducted to determine the diagnostic efficacy of the GALAD score in populations with different etiologies or from different countries.

## 2. Materials and Methods

### 2.1. Literature Search and Study Selection

Studies about the GALAD score in HCC diagnosis were searched in the *Web of Science*, *PubMed*, *Scopus*, *Ovid*, *Cochrane Library*, and *Embase* databases by 29 May 2022, as described in Appendix A. Through manual searching, no additional studies were identified that would have been missed in the six databases. Two investigators independently reviewed and screened titles and abstracts of publication. If their title or abstract did not indicate their applicability, a full review of some publications were conducted. The independent evaluation of articles was conducted for possible inclusion, and a third reviewer helped resolve any disagreements. The study was performed on the basis of Preferred Reporting Items for Systematic Review and Meta-analysis (PRISMA) guidelines and was registered in the International Prospective Register of Systematic Reviews (PROSPERO) under the registration number CRD42022329606.

### 2.2. Study Inclusion

After duplicated publication is removed, eligible studies should meet the following four inclusion criteria: (a) evaluating the diagnostic performance of the GALAD score in the HCC diagnosis; (b) setting CLD patients (excluding healthy individuals) as the control group; (c) containing critical data on parameters of diagnostic ability (e.g., sensitivity and specificity); and (d) peer-reviewed articles written in English. Meanwhile, meeting abstracts, reviews, commentary studies, and other irrelevant studies were excluded.

### 2.3. Data Extraction

By screening full texts of all the eligible articles, the following primary data were taken, including publication time, authors, study design, study location, sample size, diagnostic standard, main characteristics of subjects (age, number of males, the prevalence of cirrhosis, number of early-stage cases, and the type of control group), and diagnostic accuracy indexes (sensitivity, specificity, thresholds, and area under the receiver operating characteristic curve (AUC) values). Data extraction was conducted independently by the first and second authors and proofread by the third author.

### 2.4. Statistical Analysis

A four-grid table (including true-positive (TP), false-positive (FP), false-negative (FN), and true-negative (TN) cases) of each study cohort was obtained based on sample size, sensitivity (or TP rate), and specificity (or TN rate). First of all, the threshold effect of data was assessed to determine whether a bivariate analysis could be used [16]. Using the Q-test or I^2^ statistics, the evaluation of heterogeneity was performed. If significant heterogeneity was indicated (*p* < 0.10 and I^2^ > 50%), a random-effect model was recommended; otherwise, the fixed-effect model can be applied with evidence of a lack of heterogeneity [17]. Then, the overall diagnostic performance of the GALAD score was assessed by calculating the pooled sensitivity, pooled specificity, pooled positive likelihood ratio (PLR), pooled negative likelihood ratio (NLR), diagnostic odds ratio (DOR), estimated AUC, and their 95% confidence intervals (CI) [18]. In diagnostic studies, AUC is a critical index comprehensively weighing both sensitivity and specificity, so an AUC value close to 1.0 is considered as an excellent diagnostic accuracy, but a value of ~0.5 implies a poor efficacy. Subsequently, using subgroup analyses, we evaluated the diagnostic ability of the GALAD score for identifying early-stage HCC (defined as tumors within Barcelona clinic liver cancer (BCLC) staging system or tumors within Milan criteria) in CLD patients, discriminating HCC in patients with different etiologies (including HBV, HCV, non-viral liver disease, and cirrhosis), and differentiating HCC in patients from Western countries. The estimation of publication bias was used by the Deek’s funnel-plot asymmetry test. Using Stata software Version 17.0 and Review Manager (RevMan) Version 5.4, all analyses were conducted. *p*-value < 0.05 was considered statistically significant, and 0.05 < *p*-value < 0.10 was deemed less significant.

## 3. Results

### 3.1. Literature Search

A total of 427 studies identified by the search strategies were retrieved (*PubMed* = 27, *Embase* = 65, *Web of Science* = 53, *Cochrane* = 1, *Scopus* = 223, *Ovid* = 58). After removing 143 duplicates, 284 records were analyzed by screening titles and abstracts. Then, 263 publications (including meeting abstracts, reviews, commentary, or other irrelevant studies) were excluded. Following reading the full texts, we removed 3 studies with overlapping data [15,19,20] and 3 without required data [21,22,23], and ultimately, 15 studies were involved in the meta-analysis [24,25,26,27,28,29,30,31,32,33,34,35,36,37,38]. Figure 1 outlines the study selection process.

### 3.2. Study Characteristics

Overall, 15 articles with 19 study cohorts were published from 2016 to 2022. The characteristics of these studies evaluating the performance of the GALAD score to diagnose HCC are described in Table 1 [24,25,26,27,28,29,30,31,32,33,34,35,36,37,38]. Nearly half the study cohorts (*n* = 9) were prospective in design; 10 studies were conducted in Western countries, 5 in Eastern Asia, and 4 in multicenter from different countries. Of all the included studies, HCC diagnosis was confirmed by histological or radiological examination according to international practice guidelines, with two cohorts only by radiological examination and one only by histological examination. Of 19,021 patients, 4515 developed HCC, with a higher mean/median age than the controls. The vast majority of HCC cases occurred in males and the setting of cirrhosis. One study recorded the detailed proportions of HCC patients receiving different therapies (Appendix A) [24]. Data on the GALAD score for detecting any-stage HCC and early-stage HCC (defined as tumors within BCLC 0/A staging or tumors within Milan criteria) are shown in Table 2. The AUC values of the GALAD score for detecting any-stage HCC ranged from 0.84 to 0.98. Most study cohorts (15/19) regarded −0.63 as the cut-off value for diagnosing HCC. Among these cohorts, there were 2309 FP cases versus 13,421 control patients, with FP rates ranging from 4.2% to 27.2% (Table 2).

### 3.3. Quality Assessment

The quality assessments of all the studies were evaluated through the revised Quality Assessment Tool for Diagnostic Accuracy Studies—2 (QUADAS-2) scale. In Appendix A, there was low risk of bias in the reference standard, flow, and timing as well as low applicability concerns in the index test and reference standard; and the patient selection bias in 16 study cohorts was considered high or unclear, which was mainly attributed to the retrospective case-control design. As a whole, all the studies were of intermediate quality.

### 3.4. GALAD Score for Any-Stage HCC Detection

A threshold effect test was first performed, and there was no threshold effect. As presented in Figure 2 and Table 3, in the entire cohorts, the pooled sensitivity, specificity, and estimated AUC were 0.82 (95% CI: 0.78–0.85), 0.89 (95% CI: 0.85–0.91), and 0.92 (95% CI: 0.89–0.94), respectively. The PLR and NLR were 7.2 (95% CI: 5.5–9.4) and 0.21 (95% CI: 0.17–0.25), respectively (Appendix A). The DOR was 35 (95% CI: 24–52). A Fagan plot is shown in Appendix A; that is to say, if the pre-test probability was 24% for HCC diagnosis, the probability of having HCC was 70% in the case of a positive GALAD score and only 8% in the case of a negative result.

### 3.5. Meta-Regression

The regression analysis investigated the possible source of heterogeneity. Appendix A demonstrates that the study design, the number of study cohort, and the race of subjects might cause the difference of the sensitivity and specificity among the 19 study cohorts after univariable meta-regression (all *p* < 0.01). However, the results of multivariable meta-regression (Appendix A) indicated that no factors were sources of heterogeneity (*p* < 0.10).

### 3.6. Subgroup Analyses of GALAD Score for Early-Stage HCC Detection

Subgroup analyses of the GALAD score for diagnosing early-stage HCC were conducted (Table 3). Based on the definition of BCLC, the pooled sensitivity, pooled specificity, and estimated AUC were 0.73 (95% CI: 0.66–0.79), 0.87 (95% CI: 0.81–0.91), and 0.86 (95% CI: 0.82–0.88), respectively (Figure 3A,B). The pooled PLR, pooled NLR, and DOR were 5.4 (95% CI: 3.8–7.7), 0.31 (95% CI: 0.25–0.38), and 18 (95% CI: 12–26), respectively (Appendix A). However, only 38% of early-stage patients can be identified by AFP, with an AUC value of 0.70 (95%CI: 0.66–0.74) (Appendix A). For detecting HCC within BCLC 0/A in cirrhotic population, the GALAD score had a pooled sensitivity, pooled specificity, and estimated AUC of 0.78 (95% CI: 0.66–0.87), 0.80 (95% CI: 0.72–0.87), and 0.86 (95% CI: 0.83–0.89) (Appendix A). In contrast, if the early tumor was defined as Milan criteria, the pooled specificity and AUC value were slightly higher, but the sensitivity was lower, and the results of the GALAD score for detecting HCC within Milan criteria are shown in Figure 3C,D and Appendix A. Additionally, Appendix A shows Fagan diagrams evaluating the ability of the GALAD score for discriminating early-stage tumors.

### 3.7. Subgroup Analyses of GALAD Score for HCC Detection in Different Etiologies

Based on HCC etiologies, most studies investigated the overall diagnostic efficacy of the GALAD score for discriminating HCC in populations with HBV infection, HCV infection, non-viral liver diseases (e.g., alcoholic liver disease (ALD), nonalcoholic fatty liver disease (NAFLD), and autoimmune liver disease), or cirrhosis (Table 3, Figure 4, and Appendix A). Among the four subgroups (HBV versus HCV versus non-viral liver diseases versus cirrhosis), the higher pooled sensitivities and AUC values were observed in HCV and non-viral liver diseases subgroups (sensitivity: 0.76 (95% CI: 0.69–0.81) versus 0.86 (95% CI: 0.80–0.91) versus 0.87 (95% CI: 0.81–0.91) versus 0.76 (95% CI: 0.69–0.82); AUC: 0.85 (95% CI: 0.82–0.88) versus 0.94 (95% CI: 0.91–0.95) versus 0.94 (95% CI: 0.92–0.96) versus 0.87 (95% CI: 0.83–0.89)), but a highest pooled specificity and DOR were shown in the HBV subgroup (specificity: 0.96 (95% CI: 0.87–0.99) versus 0.89 (95% CI: 0.83–0.93) versus 0.91 (95% CI: 0.85–0.95) versus 0.85 (95% CI: 0.79–0.90); DOR: 69 (95% CI: 19–249) versus 50 (95% CI: 26–94) versus 66 (95% CI: 36–121) versus 18 (95% CI: 10–34)).

### 3.8. Subgroup Analyses of GALAD Score for HCC Detection in Different Study Designs

Different study designs might influence the diagnostic ability of the GALAD score. Appendix A shows the pooled sensitivities and specificities and summary ROC curves of the GALAD score for discriminating HCC in prospective and retrospective cohorts. In comparison, slightly higher pooled sensitivity (0.83 (95% CI: 0.77–0.88) versus 0.80 (95% CI: 0.75–0.84)), specificity (0.89 (95% CI: 0.85–0.93) versus 0.88 (95% CI: 0.82–0.92)), DOR (42 (95% CI: 24–72) versus 28 (95% CI: 17–46)), and AUC (0.93 (95% CI: 0.90–0.95) versus 0.89 (95% CI: 0.86–0.91)) values were observed in retrospective cohorts (Table 3).

### 3.9. Subgroup Analyses of GALAD Score for HCC Detection in Different Study Locations

Of 19 study cohorts, there were 10 conducted in Western countries, 5 in East Asian ones, and 4 in ones from different continents. As presented in Table 3 and Appendix A, the GALAD score detected HCC in Western countries, with a higher pooled sensitivity (0.85 (95% CI: 0.79–0.89) versus 0.76 (95% CI: 0.70–0.81) versus 0.79 (95% CI: 0.73–0.84)), higher AUC value (0.93 (95% CI: 0.91–0.95) versus 0.85 (95% CI: 0.81–0.87) versus 0.91 (95% CI: 0.89–0.94)), and higher DOR (41 (95% CI: 22–77) versus 30 (95% CI: 15–58) versus 29 (95% CI: 17–49)) compared with East Asian countries and countries from different continents.

### 3.10. Publication Bias

In order to detect publication bias, Deeks’ funnel plot asymmetry test was employed. The nearly vertical regression line and the test result (*p* = 0.52) demonstrated no publication bias (Appendix A).

## 4. Discussion

The present study demonstrated an excellent overall diagnostic performance of the GALAD score for detecting HCC, with a sensitivity, specificity, and AUC value of 0.82, 0.89, and 0.92, respectively. Notably, this HCC-specific diagnostic model can detect ~70% of early-stage tumors regardless of whether the definition is based on the BCLC system or Milan criteria. Among different groups stratified by etiologies, the pooled sensitivities approximated to 80% and specificities to 90%; in particular, the highest sensitivities and AUC values were observed among populations with HCV and non-viral liver diseases.

Generally, a patient at high risk for HCC (e.g., those with CLD) should be regularly monitored; if the screening tool indicates the disease might have progressed into HCC, diagnosis can be made through radiological imaging or biopsy. The patient receiving curative treatment at an early stage of HCC will have a good long-term prognosis. Currently, hepatic ultrasonography with or without AFP is an effective means of HCC screening in the population at risk, but there is no consensus on whether the GALAD score can serve as a better screening tool. Yang JD et al. pointed out the AUC of the GALAD score for diagnosing HCC was higher than that of ultrasonography, and the overall diagnostic performance could be improved when combining the GALAD score and ultrasonography [38]. For detecting early-stage HCC (BCLC 0/A), the AUC for GALAD+ultrasound remained high at 0.97 (95%CI: 0.95–0.99), superior to ultrasound (0.82 (95%CI: 0.76–0.87)). At the same specificity of 79%, GALAD+ultrasound had a sensitivity of >0.95 for identifying early-stage HCC, but ultrasound had 0.92; in other words, the addition of the GALAD score into ultrasound surveillance could identify more HCC patients. A previous meta-analysis indicated that ultrasonography for any-stage HCC detection had a sensitivity of 84% (similar to 82% of GALAD score) but a sensitivity of only 47% for identifying early-stage HCC (obviously lower to 65% or 73% of GALAD score in our meta-analysis) [39]. This direct and circumstantial evidence implies the robust power of GALAD score as a screening or diagnostic tool, which should be further validated by more studies with high quality.

The GALAD score is comprised of five parameters, i.e., three HCC-specific tumor biomarkers (AFP, AFP-L3, and DCP) and two demographic characteristics (age and gender). AFP, a glycoprotein, is synthesized during fetal life under physiological conditions and is reduced in expression after birth. Upregulation of AFP is commonly observed in hepatocyte regeneration, hepatocarcinogenesis, and embryonic cancers, which suggests it can function as a biomarker for diagnosing HCC [40]. However, its diagnostic ability for HCC remains under debate, with 60% sensitivity and 84% specificity at the threshold value of 20 µg/L [9]. AFP-L3 is only secreted by HCC cells, serving as a more HCC-specific marker. Its level was associated with tumor differentiation [41]. At 10% cut-off, AFP-L3%, defined as the ratio of AFP-L3 to total AFP, has 95% specificity but ~51% sensitivity for identifying HCC [40]. Normally, healthy individuals cannot detect DCP in their serum, as it is an abnormal prothrombin produced by malignant hepatocytes; its abnormal secretion seems to be attributed to an acquired defect in the posttranslational vitamin-K-dependent carboxylation of a prothrombin precursor, so it can be utilized as an HCC-specific biomarker, with a sensitivity of ~60% and a specificity of ~90% [40]. In addition, some demographic characteristics are also closely related to HCC. For example, there is a strong link between HCC and old age; and high prevalence in men is also reported [2]. The combination of the three HCC-specific biomarkers and two risk factors produced a synergistic effect, with an improved overall diagnostic accuracy. Despite the emergence of multiple novel biomarkers identified for detecting HCC in recent years, these five indexes are relatively affordable and easy to obtain frequently.

The development of the GALAD score was based on the data from a British cohort, where HCV, ALD, and NAFLD are the main pathogenic factors for HCC [2,15]. As shown in Table 3, the GALAD score yielded a higher pooled sensitivity and AUC value for identifying HCC in the Western countries compared with the East Asian ones. Moreover, given significant variations in HCC etiologies between different regions (HBV in China; HCV, NAFLD, and ALD in Western countries), higher pooled sensitivities and AUC values were observed in HCV (sensitivity = 0.86; AUC = 0.94) and non-viral liver diseases (sensitivity = 0.87; AUC = 0.94) populations; by contrast, the GALAD score yielded a lowest sensitivity (0.76) and AUC (0.85) in the HBV population, which means the GALAD score could detect 10% more HCC patients in those with HCV or non-viral liver diseases. On the other hand, compared with ultrasound or other diagnostic models constructed based on hepatitis B patients (e.g., AGED, REACH-B, PAGE-B, and mPAGE-B), GALAD still had a higher AUC value for detecting HBV-related HCC [37,38]. Of note, the predominant etiologies causing HCC are changing currently, and growing data indicate that it is becoming more common in NAFLD to develop HCC [42]. To our knowledge, there are few studies on the efficacy of the above three biomarkers or the GALAD score for detecting NAFLD-associated HCC [25,43,44]. Best J et al. investigated the ability of the GALAD score for detecting early-stage HCC in NASH, demonstrating a superior diagnostic performance of the GALAD score with an AUC value of 0.94 (95%CI: 0.86–1.03) in a noncirrhotic population and of 0.85 (95%CI: 0.73–0.97) in a cirrhotic one. At a cut-off value of −0.63, GALAD could identify 85.7% of early-stage HCC in patients with noncirrhotic NASH [25]. Another study conducted by Lambrecht J et al. evaluated its diagnostic performance for identifying HCC in cirrhotic NAFLD, with an AUC value of 0.91 (95%CI: 0.83–0.98) [30]. Given the different pathogenesis of HCC caused by various etiologies, the diagnostic ability of the GALAD score for detecting HCC in patients with ALD or with multiple etiologies is not clearly established. In different populations, how to give full play to the optimal performance of the model and whether it needs further optimization are also worth investigating.

Since the emergence of the GALAD score, several diagnostic models for HCC have been developed based on the native population, such as ASAP (including AFP, DCP, age, and gender) [22], male-ABCD (including age, γ-glutamyl-transpeptidase, platelets, white blood cells, DCP, and AFP) [37], GALAD-C (including gender, age, AFP-L3%, AFP, and DCP) [32], GAAP (including AFP, DCP, age, and gender) [32], and HES (including rate of AFP change, AFP, alanine aminotransferase, platelets, and age) [45]. The combination of tumor biomarkers and high-risk factors is the largest common denominator across these novel algorithms. Given the current limited evidence, there is no conclusive confirmation which one has the optimal diagnostic ability. On the other hand, despite the globally wide utilization of AFP, no introduction of DCP and/or AFP-L3 has been made into clinical practice in some countries and regions. Furthermore, now, there are no uniform opinions on the threshold values of these novel models for detecting any-stage or early-stage HCC. As a continuous variable, the different cut-off values of a diagnostic model can directly change the numbers of TP, FP, FN, and TN cases; that is to say, in a fixed study cohort, a sensitivity and FP rate will decrease with the increase of cut-off value. In most articles analyzed in the current meta-analysis, the cut-off value of GALAD score was set as −0.63, which was an optimal threshold based on the British data. However, the best value of the GALAD score is always diverse in different populations or in detecting early-stage HCC. As such, these restrictions cause a limitation in generalizing the algorithms to a certain degree.

Generally, the diagnosis of a solid tumor is confirmed by histological analysis; of all solid tumors, only HCC can also be diagnosed by clinical diagnostic criteria, which is recognized at home and abroad. In all the included studies, HCC subjects were diagnosed by histological or radiological examination according to international practice guidelines as the gold standard for evaluating the diagnostic performance of GALAD score. Without detailed data on diagnostic modality of any patient, minor differences in diagnostic gold standards among the different studies may simultaneously change the numbers of TP and FP or those of FN and TN. Similarly, in the majority of studies, some CLD patients did not receive further follow-up or were not retested for HCC using “gold-standard” tests; delayed recognition of HCC cases in the so-called control group would also mistake the two sets of values. These might lead to incorrect estimation of GALAD score performance

Although the pooled sensitivities, specificities, and AUC values of GALAD score for any-stage HCC detection were high, wide variation in its efficacy was found among the included articles. As such, meta-regression was conducted to explore the heterogeneity, but some factors (including sample size, race, and study design) were not capable of fully explaining the differences in GALAD score performance. Insufficient information in the included studies made it difficult to determine the impact of several other factors (e.g., detection equipment for biomarkers, enrollment time of subjects, and the ratio of etiologies). Furthermore, subgroup analyses also demonstrated a good diagnostic ability of GALAD for identifying early-stage HCC or differentiating HCC in different CLD.

For research purposes, parameters before or at diagnosis, such as age, gender, etiologies, tumor biomarkers, and tumor characteristics, are collected for analysis in most clinical diagnostic studies. Without further follow-up, few parameters after diagnosis are recorded in studies, including therapeutic modalities. Of all the included studies, only one study recorded the detailed proportions of HCC patients receiving different therapies. According to current international guidelines, surgery is recommended as an optimal therapy for early-stage HCC and non-surgical treatment for intermediate-stage or advanced-stage HCC. In other words, the distribution of tumor stages, as significant parameters usually described in diagnostic studies, may indirectly reflect the proportions of HCC patients receiving surgical or non-surgical treatment; and the performance evaluation of the GALAD score for detecting HCC within BCLC 0/A staging or Milan criteria could indirectly indicate its ability for discriminating HCC in patients undergoing surgery. Undoubtedly, the diagnostic performance of GALAD score for detecting early-stage HCC is more valuable for e assessment compared with its ability for advanced-stage tumors. Our subgroup analyses demonstrated a good diagnostic accuracy of the GALAD score for identifying BCLC 0/A HCC, with a pooled sensitivity, specificity, and AUC of 0.73 (95% CI: 0.66–0.79), 0.87 (95% CI: 0.81–0.91), and 0.86 (95% CI: 0.82–0.88), respectively; and based on the definition of Milan criteria, the pooled sensitivity, pooled specificity, and estimated AUC were 0.65 (95% CI: 0.56–0.72), 0.91 (95% CI: 0.87–0.94), and 0.87 (95% CI: 0.83–0.89), respectively. These means that a majority of HCC patients can be identified in a timely way at their early stages by the GALAD score.

To the best of our knowledge, no meta-analysis that comprehensively evaluates the overall performance of GALAD score for diagnosing HCC has been conducted before. However, interpretation of our results must take into account the limitations of the articles included. First, 15 studies were mainly performed in North America, Asia, and Europe, but studies in South America, Australia, and Africa are lacking; further investigation of its performance in these regions is needed. Second, in the majority of studies, CLD patients under surveillance tests did not receive further follow-up or were not retested for HCC using a so-called “gold-standard” test, which might lead to mistaken estimation of GALAD score performance. Third, differences in diagnostic gold standards among the different studies may also lead to some inaccuracies in performance evaluation. Fourth, pooled results may be influenced by some studies that had high bias risks. Fifth, some meaningful information (e.g., proportions of HCC patients receiving different therapies and data on GALAD for discriminating HCC in patients undergoing surgery) cannot be extracted from the majority of studies, making it difficult to evaluate its diagnostic performance from different angles.

To conclude, we demonstrated the superior diagnostic accuracy of GALAD score for detecting HCC with a high sensitivity and specificity, especially for BCLC-0/A HCC, with a relatively stable diagnostic performance. The GALAD score promisingly serves as a complement to HCC screening or an alternative surveillance strategy.

## Figures and Tables

**Figure 1 jcm-12-00949-f001:**
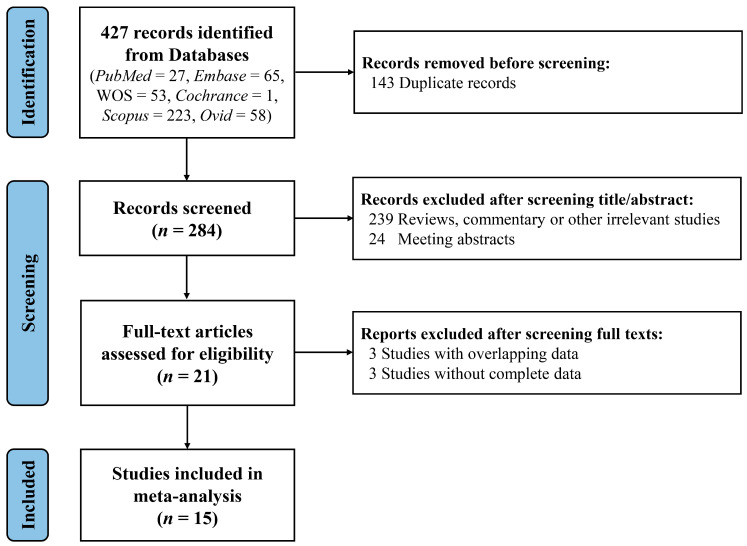
The flow chart of studies included in the meta-analysis. WOS, *Web of Science*.

**Figure 2 jcm-12-00949-f002:**
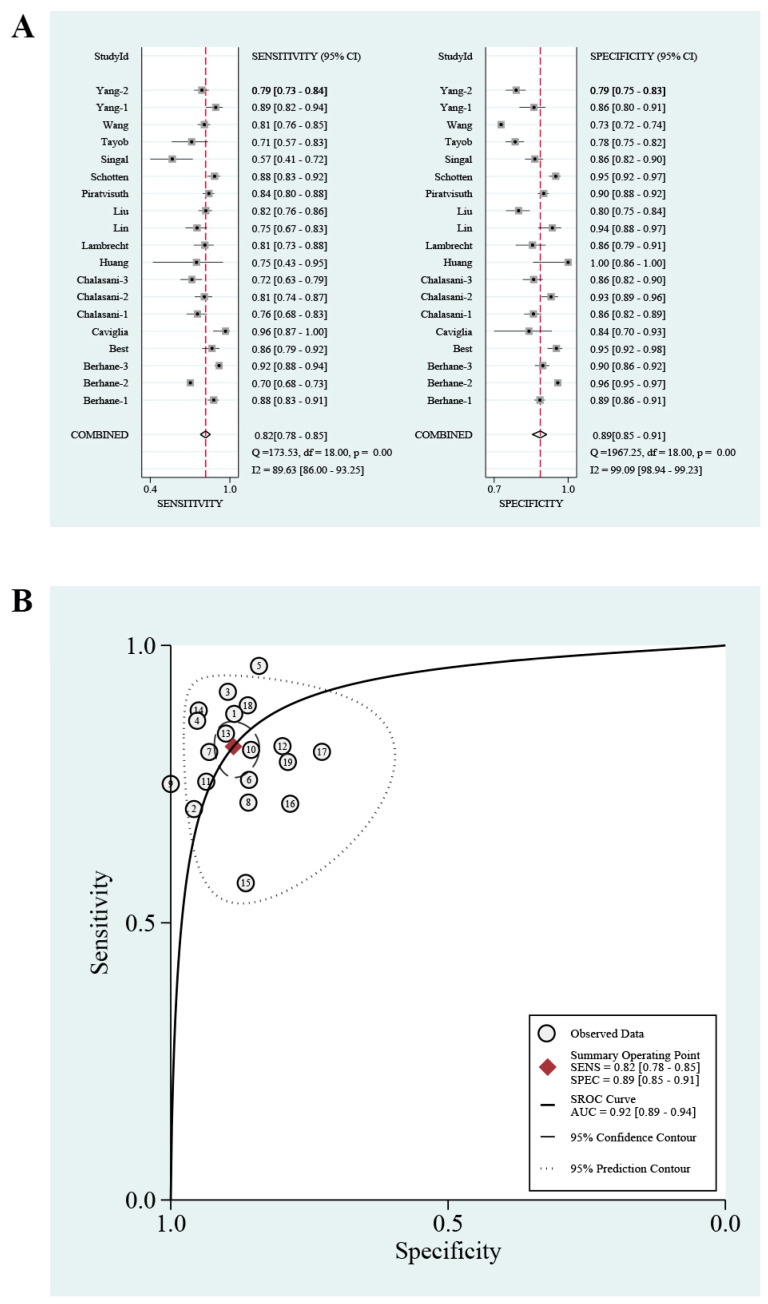
The pooled sensitivity and specificity with Forest plots (**A**) and summary receiver operating characteristic curve (**B**) of GALAD score for discriminating any-stage hepatocellular carcinoma in chronic liver diseases.

**Figure 3 jcm-12-00949-f003:**
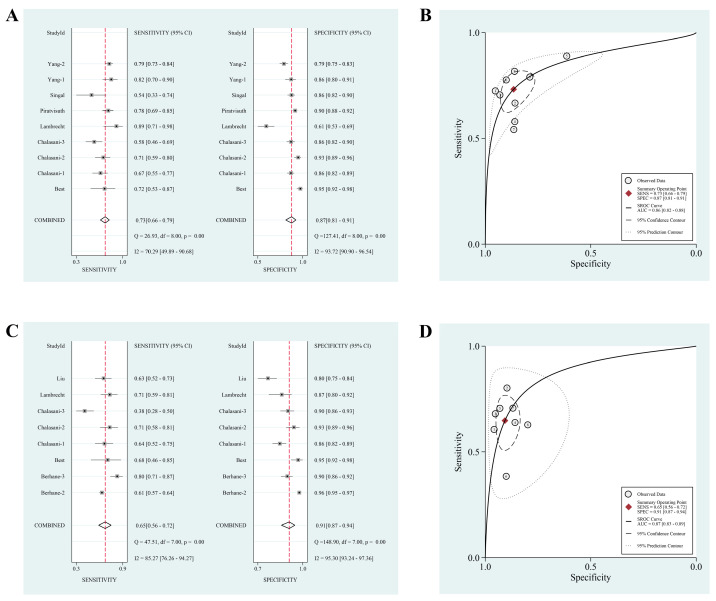
The pooled sensitivity and specificity with Forest plots as well as summary receiver operating characteristic curves of GALAD score for discriminating early-stage hepatocellular carcinoma within Barcelona Clinic Liver Cancer 0/A staging (**A**,**B**) or within Milan criteria (**C**,**D**) in chronic liver diseases.

**Figure 4 jcm-12-00949-f004:**
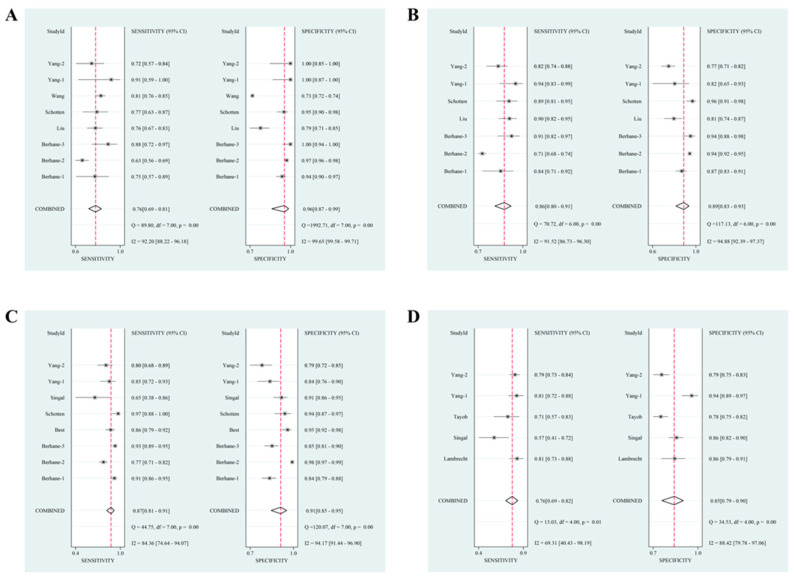
The pooled sensitivity specificity with Forest plots of GALAD score for discriminating hepatocellular carcinoma in patients with different etiologies. (**A**) Hepatitis B virus; (**B**) hepatitis C virus; (**C**) non-viral liver diseases; (**D**) cirrhosis.

**Table 1 jcm-12-00949-t001:** Characteristics of studies.

				HCC					Control				
Author	Publication Time	Study Location	Study Design	Number	Age, Years	Male, *n* (%)	Cirrhosis, *n* (%)	Diagnostic Modality	Control Type	Number	Age, Years	Male, *n* (%)	Cirrhosis, *n* (%)
Berhane-1 [24]	2016	Germany	Retrospective	275	65.4 †	231 (84.0)	\	Histological or radiological	CLD	1003	50.5 †	500 (55.4)	\
Berhane-2 [24]	2016	Japan	Retrospective	1514	67.8 †	1079 (71.3)	\	Histological or radiological	CLD	2962	61.0 †	1422 (48.0)	\
Berhane-3 [24]	2016	United Kingdom	Retrospective	394	65.8 †	325 (82.5)	\	Histological or radiological	CLD	439	54.9 †	256 (58.3)	\
Best [25]	2020	Germany	Retrospective	125	70.5 ‡	84 (67.2)	95 (76.0)	Histological or radiological	NASH	231	52.0 ‡	120 (51.9)	49 (20.9)
Caviglia [26]	2016	Italy	Prospective	54	69.5 †	45 (83.3)	52 (96.3)	Radiological	CLD	44	53.2 †	23 (52.3)	14 (31.8)
Chalasani-1 [27]	2021	United States, Europe, and Asia	Prospective	136	64.0 ‡	96 (70.6)	130 (95.6)	Histological or radiological	CLD	404	64.0 ‡	234 (57.9)	374 (92.6)
Chalasani-2 [27]	2021	United States, Europe, and Asia	Prospective	156	63.0 ‡	130 (83.3)	151 (96.8)	Histological or radiological	CLD	245	59.0 ‡	144 (58.8)	226 (92.2)
Chalasani-3 [28]	2021	United States, Europe, and Asia	Prospective	135	64.0 ‡	103 (76.3)	120 (89.6)	Histological or radiological	CLD	302	64.0 ‡	168 (55.6)	264 (87.4)
Huang [29]	2022	China	Prospective	12	52.1†	10 (83.33)	\	Histological	CLD	24	\	\	\
Lambrecht [30]	2021	Germany	Retrospective	122	66.0 ‡	95 (77.9)	122 (100)	Histological or radiological	CLD	145	54.0 ‡	99 (68.3)	145 (100.0)
Lin [31]	2022	China	Prospective	122	55.0 ‡	106 (86.9)	45 (36.9)	Histological or radiological	CLD	125	47.0 ‡	83 (66.4)	46 (36.8)
Liu [32]	2020	China	Retrospective	242	59.0 ‡	176 (72.7)	\	Histological or radiological	CLD	283	52.0 ‡	167 (59.0)	187 (64.7)
Piratvisuth [33]	2021	China, Germany, Spain, and Thailand	Prospective	308	60.8 †	243 (78.9)	239 (77.6)	Histological or radiological	CLD	740	55.5 †	417 (56.4)	395 (53.4)
Schotten [34]	2021	Germany	Retrospective	196	\	\	\	Histological or radiological	CLD	377	\	\	\
Singal [35]	2022	United States	Retrospective	42	53.5 ‡	30 (71.4)	42 (100.0)	Histological or radiological	CLD	355	52.0 ‡	208 (58.6)	355 (100.0)
Tayob [36]	2022	United States	Prospective	50	64.7 †	50 (100.0)	50 (100.0)	Histological or radiological	CLD	484	63.0 †	472 (97.5)	484 (100.0)
Wang [37]	2021	China	Prospective	302	\	302 (100)	\	Radiological	HBV	5851	\	5851 (100)	\
Yang-1 [38]	2019	United States	Retrospective	111	63.9 †	86 (77.5)	109 (98.2)	Histological or radiological	CLD	180	57.1 †	96 (53.3)	154 (85.6)
Yang-2 [38]	2019	United States	Retrospective	233	60.8 †	172 (73.8)	233 (100.0)	Histological or radiological	CLD	412	54.9 †	287 (69.7)	412 (100.0)

CLD, chronic liver disease; HBV, hepatitis B virus; HCC, hepatocellular carcinoma; NASH, nonalcoholic steatohepatitis. † mean age; ‡ median age.

**Table 2 jcm-12-00949-t002:** Summary of the diagnostic performance of the GALAD score for detecting any-stage HCC or early-stage HCC.

Author	TP	FP	FN	TN	AUC	Cut-Off Value	Sensitivity, %	Specificity, %
Any-stage HCC								
Berhane-1 [24]	241	103	34	797	0.94	−0.63	87.6	88.6
Berhane-2 [24]	1067	124	447	2838	0.93	−0.63	70.5	95.8
Berhane-3 [24]	350	45	32	392	0.97	−0.63	91.6	89.7
Best [25]	108	11	17	220	0.96	−0.63	86.4	95.2
Caviglia [26]	52	7	2	37	0.98	−2.59	96.3	84.1
Chalasani-1 [27]	103	57	33	347	0.88	−0.63	76.0	86.0
Chalasani-2 [27]	126	17	30	228	0.92	−0.63	81.0	93.0
Chalasani-3 [28]	96	42	38	258	0.87	−0.63	72.0	86.0
Huang [29]	9	0	3	24	0.85	−0.07	75.0	100.0
Lambrecht [30]	99	21	23	124	0.90	−0.63	81.2	85.5
Lin [31]	92	8	30	117	0.90	−0.63	75.4	93.6
Liu [32]	198	57	44	226	0.89	0.95	81.8	79.9
Piratvisuth [33]	259	73	49	661	0.95	\	84.1	90.0
Schotten [34]	173	19	23	358	0.97	−0.63	88.3	95.0
Singal [35]	24	48	18	307	0.79	−0.63	57.1	86.5
Tayob [36]	35	107	14	390	0.84	−0.63	71.4	78.5
Wang [37]	244	1595	58	4256	0.84	−0.63	80.8	72.8
Yang-1 [38]	99	25	12	155	0.95	−0.63	89.0	86.0
Yang-2 [38]	184	87	49	325	0.88	−0.63	79.0	79.0
HCC within BCLC 0/A staging								
Best [25]	21	11	8	220	0.92	−0.63	72.4	95.2
Chalasani-1 [27]	54	57	27	347	0.83	−0.63	67.0	86.0
Chalasani-2 [27]	55	17	23	228	0.89	−0.63	71.0	93.0
Chalasani-3 [28]	44	42	32	258	0.81	−0.63	58.0	86.0
Lambrecht [30]	24	56	3	89	0.81	−2.286	88.9	61.6
Piratvisuth [33]	97	73	28	661	0.91	\	77.7	90.0
Singal [35]	13	48	11	307	0.78	−0.63	53.8	86.5
Yang-1 [38]	49	25	11	155	0.92	−0.63	82.0	86.0
Yang-2 [38]	184	87	49	325	0.88	−0.63	79.0	79.0
HCC within Milan criteria								
Berhane-2 [24]	538	124	350	2838	0.91	−0.63	60.6	95.8
Berhane-3 [24]	85	45	21	392	0.93	−0.63	80.2	89.7
Best [25]	17	11	8	220	0.91	−0.63	68.0	95.2
Chalasani-1 [27]	46	57	26	347	\	−0.63	64.0	86.0
Chalasani-2 [27]	48	17	20	228	\	−0.63	71.0	93.0
Chalasani-3 [28]	30	30	48	272	\	−0.182	39.0	90.0
Lambrecht [30]	53	19	22	126	0.86	−0.4405	71.1	87.0
Liu [32]	54	57	32	226	0.81	0.946	62.8	79.9

AUC, area under the curve; BCLC, Barcelona Clinic Liver Cancer; FN, false-negative cases; FP, false-positive cases; HCC, hepatocellular carcinoma; TN, true-negative cases; TP, true-positive cases.

**Table 3 jcm-12-00949-t003:** Statistical analyses of GALAD score for discriminating HCC in different groups.

Group	Number of Study Cohorts	Number HCC	Number Control	Pooled Sensitivity (95%CI)	Pooled Specificity (95%CI)	Pooled PLR (95%CI)	Pooled NLR (95%CI)	DOR (95%CI)	AUC (95%CI)
Entirety									
Any-stage HCC	19	4515	14,506	0.82 (0.78–0.85)	0.89 (0.85–0.91)	7.2 (5.5–9.4)	0.21 (0.17–0.25)	35 (24–52)	0.92 (0.89–0.94)
Early stage									
Within BCLC 0/A	9	733	3006	0.73 (0.66–0.79)	0.87 (0.81–0.91)	5.4 (3.8–7.7)	0.31 (0.25–0.38)	18 (12–26)	0.86 (0.82–0.88)
Within Milan criteria	8	1398	5009	0.65 (0.56–0.72)	0.91 (0.87–0.94)	6.9 (4.7–10.2)	0.39 (0.31–0.49)	18 (11–30)	0.87 (0.83–0.89)
Etiologies									
HBV	8	841	7145	0.76 (0.69–0.81)	0.96 (0.87–0.99)	17.5 (5.3–57.2)	0.25 (0.19–0.33)	69 (19–249)	0.85 (0.82–0.88)
HCV	7	1517	2225	0.86 (0.80–0.91)	0.89 (0.83–0.93)	7.7 (4.9–12.2)	0.16 (0.11–0.23)	50 (26–94)	0.94 (0.91–0.95)
Non-viral liver diseases	8	992	2216	0.87 (0.81–0.91)	0.91 (0.85–0.95)	9.5 (5.7–15.9)	0.15 (0.10–0.21)	66 (36–121)	0.94 (0.92–0.96)
Cirrhosis	5	555	1563	0.76 (0.69–0.82)	0.85 (0.79–0.90)	5.2 (3.5–7.7)	0.28 (0.21–0.37)	18 (10–34)	0.87 (0.83–0.89)
Study design									
Prospective	9	1273	8224	0.80 (0.75–0.84)	0.88 (0.82–0.92)	6.4 (4.4–9.3)	0.23 (0.19–0.29)	28 (17–46)	0.89 (0.86–0.91)
Retrospective	10	3242	6282	0.83 (0.77–0.88)	0.89 (0.85–0.93)	7.9 (5.5–11.3)	0.19 (0.14–0.26)	42 (24–72)	0.93 (0.90–0.95)
Study location									
Western countries	10	1589	3578	0.85 (0.79–0.89)	0.88 (0.84–0.91)	7.1 (5.1–9.8)	0.17 (0.12–0.25)	41 (22–77)	0.93 (0.91–0.95)
East Asian countries	5	2192	9245	0.76 (0.70–0.81)	0.91 (0.79–0.96)	8.0 (3.7–17.2)	0.27 (0.23–0.31)	30 (15–58)	0.85 (0.81–0.87)
Countries from different continents	4	734	1683	0.79 (0.73–0.84)	0.89 (0.86–0.91)	6.9 (5.2–9.2)	0.24 (0.18–0.31)	29 (17–49)	0.91 (0.89–0.94)

AUC, area under the curve; BCLC, Barcelona clinic liver cancer; CI, confidence intervals; DOR, diagnostic odds ratio; HBV, hepatitis B virus; HCC, hepatocellular carcinoma; HCV, hepatitis C virus; NLR, negative likelihood ratio; PLR, positive likelihood ratio.

## Data Availability

Detailed data supporting the findings of this study can be accessed in the article or its Appendix A, and additional data not presented in the article can be available from the corresponding author upon reasonable request.

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
