# Peer review of "The Performance of GALAD Score for Diagnosing Hepatocellular Carcinoma in Patients with Chronic Liver Diseases: A Systematic Review and Meta-Analysis"

_jcm, 2023, doi:10.3390/jcm12030949_

Round 1

Reviewer 1 Report

In this meta-analysis, the Authors investigated the overall diagnostic performance in differentiating HCC from chronic liver diseases by GALAD score.

There are some concerns I have about the methodology.

Material and methods

-Let us begin by explaining how the true positive [TP], false positive [FP], false negative [FN], and true negative [TN] were considered in relation to which of the different gold standards used to diagnose HCC. In other word specify about the pooled accuracy how was considered the gold standard for diagnosing HCC in different studies considered: simple clinical diagnosis as defined as tumor within Barcelona clinic liver cancer staging system or tumor within Milan criteria? biopsy histology? surgical specimen? It is crucial to specify the gold standard used for every study considered in your metanalysis and to discuss any difference found among studies.

-Data Extraction: in the studies considered for matanalysis I suppose that were included patients submitted to surgical procedures (liver resection, liver transplantation) or different multimodal therapies; could you specify in which proportion patients received surgical or non-surgical procedure or both?

-AUC: AUC represents the degree or measure of separability. It tells how much the model is capable of distinguishing between classes. Higher the AUC, the better the model is at predicting accuracy classes. In a Forest plot, would it be possible to visualize a second ROC discriminating for HCC in patients undergoing surgery? Surgery is usually performed for patients in an earlier stage of HCC (Milan criteria or BCLC guidelines), where accuracy is obviously more important than accuracy in the advanced stage. This point needs to be clarified and discussed.

-It would be interesting to know how many false positives were found in different studies considered in your analysis.

Author Response

Response to Reviewer's Comments

Point 1: Let us begin by explaining how the true positive [TP], false positive [FP], false negative [FN], and true negative [TN] were considered in relation to which of the different gold standards used to diagnose HCC. In other word specify about the pooled accuracy how was considered the gold standard for diagnosing HCC in different studies considered: simple clinical diagnosis as defined as tumor within Barcelona clinic liver cancer staging system or tumor within Milan criteria? biopsy histology? surgical specimen? It is crucial to specify the gold standard used for every study considered in your metanalysis and to discuss any difference found among studies.

Response 1: Thanks for Reviewer 1’s nice and favorable comments. We also approve of the opinion that the gold standard for diagnosing HCC might affect the pooled accuracy. In most studies, HCC diagnosis was confirmed by histological or radiological examination. And a new column of data about diagnostic modality for HCC has been added into Table 1.

“In most cohorts (n = 16) , HCC diagnosis was confirmed by histological or radiological examination.” Line 140-141

Point 2: Data Extraction: in the studies considered for matanalysis I suppose that were included patients submitted to surgical procedures (liver resection, liver transplantation) or different multimodal therapies; could you specify in which proportion patients received surgical or non-surgical procedure or both?

Response 2: Thanks. The proportion of HCC patients receiving different therapies was a piece of meaningful data. After carefully reviewing all the included studies, we, unfortunately, found that only 1 study recorded the detailed proportions of HCC patients receiving different therapies, but no related data were published in the remaining 14 studies.

“1 study recorded the detailed proportions of HCC patients receiving different thera-pies (Supplementary Table S2) [24].” Line 143-145

Table S2. Proportion of HCC Patients Receiving Different Therapies.

Berhane-124

Berhane-224

Berhane-324

Transplantation, %

3.6

0

4.8

Resection, %

4.2

32.9

2.6

Ablative, %

3.0

26.9

10.5

TACE, %

26.2

21.0

38.0

Sorafenib/chemotherapy, %

14.3

1.1

16.0

Supportive, %

14.3

13.5

22.7

Other palliative, %

34.5

4.6

5.4

Point 3: AUC: AUC represents the degree or measure of separability. It tells how much the model is capable of distinguishing between classes. Higher the AUC, the better the model is at predicting accuracy classes. In a Forest plot, would it be possible to visualize a second ROC discriminating for HCC in patients undergoing surgery? Surgery is usually performed for patients in an earlier stage of HCC (Milan criteria or BCLC guidelines), where accuracy is obviously more important than accuracy in the advanced stage. This point needs to be clarified and discussed.

Response 3: Thanks for the comments. The diagnostic ability of a novel diagnostic tool should be validated in different groups. As such, We have extracted all the valuable data from the studies and investigated the performance of GALAD score for detecting any-stage HCC, early-stage HCC, HBV-HCC, HCV-HCC, and cirrhotic HCC. But no data on GALAD for discriminating HCC in patients undergoing surgery were published in all the included studies.

Fourth, some meaningful information (e.g., data on GALAD for discriminating HCC in patients undergoing surgery) cannot be extracted from the included studies, making it difficult to evaluate its performance from different angles.” Line 387-389

Point 4: It would be interesting to know how many false positives were found in different studies considered in your analysis.

Response 4: Thanks. GALAD is a model comprising five clinical parameters, i.e., gender, age, AFP-L3%, AFP, and DCP. Previous studies revealed the close relationship between the levels of these three biomarkers with several tumor-associated factors including tumor size and etiologies. The sensitivities of AFP-L3%, AFP, and DCP for detecting HCC with different tumor stages or etiologies vary. Especially, their sensitivities usually decreased in detecting early-stage HCC; in other words false positive cases would increase. In addition, the different cut-off values can change false positive cases. We have added a paragraph to discuss it in Discussion Section.

The diagnostic performance of GALAD score is strongly related to that of AFP-L3%, AFP, or DCP. Previous studies revealed the close relationship between the levels of these three biomarkers with several tumor-associated factors including tumor size and etiologies [40, 46]. The sensitivities of AFP-L3%, AFP, and DCP for detecting HCC with different tumor stages or etiologies vary. Especially, their sensitivities usually de-creased in detecting early-stage HCC; in other words false positive cases would in-crease. In addition, the different cut-off values can change false positive cases. As such, the sensitivity of GALAD can be improved by adjusting the threshold, optimizing the model (e.g., the addition of ultrasound), and so on.” Line 360-368

46 Marrero, J. A.; Feng, Z.; Wang, Y.; Nguyen, M.H.; Befeler, A.S.; Roberts, L.R.; Reddy, K.R.; Harnois, D.; Llovet, J.M.; Normolle, D.; Dalhgren, J.; Chia, D.; Lok, A.S.; Wagner, P.D.; Srivastava, S.; Schwartz, M. Alpha-fetoprotein, des-gamma carboxyprothrombin, and lectin-bound alpha-fetoprotein in early hepatocellular carcinoma. Gastroenterology. 2009, 137(1), 110–118.” Line 523-526

Reviewer 2 Report

This is a meta-analysis on the diagnostic performance of the GALAD score for detecting HCC in patients with chronic liver diseases. In principle, a meta-analysis is justified on this topic. However, the current analysis in not very helpful for the clinical management, as important take-home messages are lacking.

Major 1: The meta-analysis mixes the control populations – chronic liver disease without cirrhosis (here, NAFLD would be most important) vs. chronic liver diseases with cirrhosis (here, the added value over current surveillance strategies such as 6-month ultrasound would be most important). I understand that the data are difficult to extract from the studies, but at present very different studies are mixed. I suggest to add the following analyses: (a) Analysis from Fig. 3 (early detection rate) with only cirrhosis or non-cirrhosis controls. (b) Analysis, how many cases of early stage HCC (NNT) would be identified in a cirrhosis population, if GALAD score was added to the ultrasound surveillance. (c) Analysis, how many cases of early stage HCC (NNT) would be identified in a non-cirrhosis high-risk NAFLD population (e.g. TE > 8/12 or FIB-4 >2.6), if GALAD score was implemented in clinical routine. (d) How does the NNT change in both scenarios, if AFP would be done without the (expensive) GALAD score?

Major 2: I did not understand the conclusion regarding HBV-related HCC. I understand that the current studies do not include a large number of HBV-related HCC, but I was wondering to which extent GALAD score would be useful (compared to other risk scores like PAGE-B or 6-months-ultrasound).

Minor: the quality if the figures is very poor and must be improved.

Author Response

Response to Reviewer's Comments

Point 1: The meta-analysis mixes the control populations – chronic liver disease without cirrhosis (here, NAFLD would be most important) vs. chronic liver diseases with cirrhosis (here, the added value over current surveillance strategies such as 6-month ultrasound would be most important). I understand that the data are difficult to extract from the studies, but at present very different studies are mixed. I suggest to add the following analyses: (a) Analysis from Fig. 3 (early detection rate) with only cirrhosis or non-cirrhosis controls.

Response 1: Thanks for Reviewer 2’s nice comments. The diagnostic performance of GALAD score for detecting HCC varies in population with different etiologies, e.g., cirrhosis versus non-cirrhosis. It is very valuable to assess its overall ability for detecting early-stage HCC in cirrhotic or noncirrhotic populations. After reviewing all the included studies, we found most of the subjects included in the study had a background of cirrhosis, and the data on non-cirrhosis were few. As such, we investigated the performance of GALAD score for detecting early-stage HCC in cirrhotic population, and described it in Results Section.

For detecting HCC within BCLC 0/A in cirrhotic population, GALAD score had a pooled sensitivity, pooled specificity, and estimated AUC of 0.78 (95% CI: 0.66 - 0.87), 0.80 (95% CI: 0.72 - 0.87), and 0.86 (95% CI: 0.83 - 0.89) (Supplementary Table S6).” Line 214-217

Table S6. GALAD Score for Discriminating Hepatocellular Carcinoma within Barcelona Clinic Liver Cancer 0/A Staging in Cirrhotic Population.

Number of study cohorts

Number HCC

Number control

Pooled Sensitivity (95%CI)

Pooled Specificity (95%CI)

Pooled PLR (95%CI)

Pooled NLR (95%CI)

DOR (95%CI)

AUC (95%CI)

5

365

1113

0.78

(0.66-0.87)

0.80

(0.72-0.87)

4.0

(3.0-5.2)

0.27

(0.18-0.41)

15

(10-20)

0.86

(0.83-0.89)

Point 2: Analysis, how many cases of early stage HCC (NNT) would be identified in a cirrhosis population, if GALAD score was added to the ultrasound surveillance.

Response 2: Thanks. Professor Yang JD et al. evaluated whether the combination of liver ultrasound and GALAD score further enhances the performance of HCC detection compared to the performance of either test alone. Combining GALAD and liver ultrasound significantly improved the performance of the novel model GALADUS. For detecting early-stage HCC (BCLC 0/A), the AUC for GALADUS score remained high at 0.97 (95%CI: 0.95 - 0.99), superior to ultrasound (0.82 [95%CI: 0.76 - 0.87]). At the same specificity of 79%, GALADUS had a sensitivity of > 0.95 for identifying early-stage HCC, but ultrasound had 0.92; in other words the addition of GALAD score into ultrasound surveillance could identify more HCC patients. The related comments have been described in the Discussion Section of our manuscript.

For detecting early-stage HCC (BCLC 0/A), the AUC for GALAD+ultrasound remained high at 0.97 (95%CI: 0.95 - 0.99), superior to ultrasound (0.82 [95%CI: 0.76 - 0.87]). At the same specificity of 79%, GALAD+ultrasound had a sensitivity of > 0.95 for identifying early-stage HCC, but ul-trasound had 0.92; in other words, the addition of GALAD score into ultrasound sur-veillance could identify more HCC patients.” Line 284-289

Point 3: Analysis, how many cases of early stage HCC (NNT) would be identified in a non-cirrhosis high-risk NAFLD population (e.g. TE > 8/12 or FIB-4 >2.6), if GALAD score was implemented in clinical routine.

Response 3: Thanks for the comments. Cumulative evidence suggests that the incidence of HCC is increasing in individuals with NAFLD, which is projected to become the dominant cause of HCC globally. Different from other etiologies, a high proportion of HCC cases caused by NAFLD develop in the setting of non-cirrhotic liver. As such, it is meaningful to assess the ability of GALAD score for identifying early-stage HCC in non-cirrhosis NAFLD population. To our knowledge, there are few studies on the efficacy of GALAD score for detecting NAFLD-associated HCC. Of all the included studies, only two did. And we described the related findings of these 2 studies in the Discussion Section.

Best J et al. investigated the ability of GALAD score for detecting early-stage HCC in NASH, demonstrating a superior diagnostic performance of GALAD score with an AUC value of 0.94 (95%CI: 0.86 - 1.03) in noncirrhotic population and of 0.85 (95%CI: 0.73 - 0.97) in cirrhotic one. At a cut-off value of -0.63, GALAD could identify 85.7% of early-stage HCC in patients with noncirrhotic NASH [25]. Another study conducted by Lambrecht J et al. evaluated its diagnostic performance for identifying HCC in cirrhotic NAFLD, with an AUC value of 0.91 (95%CI: 0.83 - 0.98) [30].” Line 332-338

Point 4: How does the NNT change in both scenarios, if AFP would be done without the (expensive) GALAD score?

Response 4: Thanks. Of all the included studies, only a few assessed the diagnostic ability of AFP for detecting early-stage HCC, but we extracted the related data to evaluate its performance. As shown in Supplementary Table S5, the pooled sensitivity and AUC of AFP for BCLC 0/A HCC was 0.38 (95%CI: 0.28-0.49) and 0.70 (95%CI: 0.66-0.74), lower than those of GALAD score. And these findings were added into Results Section.

But only 38% of early-stage patients can be identified by AFP, with an AUC value of 0.70 (95%CI: 0.66 - 0.74) (Supplementary Table S5).” Line 213-214

Table S5. Alpha-fetoprotein for Detecting Hepatocellular Carcinoma within Barcelona Clinic Liver Cancer 0/A Staging.

Number of study cohorts

Number HCC

Number control

Pooled Sensitivity (95%CI)

Pooled Specificity (95%CI)

Pooled PLR (95%CI)

Pooled NLR (95%CI)

DOR (95%CI)

AUC (95%CI)

5

384

2038

0.38

(0.28-0.49)

0.97

(0.92-0.99)

13.4

(5.3-33.6)

0.64

(0.54-0.75)

21

(8-52)

0.70

(0.66-0.74)

Point 5: I did not understand the conclusion regarding HBV-related HCC. I understand that the current studies do not include a large number of HBV-related HCC, but I was wondering to which extent GALAD score would be useful (compared to other risk scores like PAGE-B or 6-months-ultrasound).

Response 5: Thanks for the comments. Similar to AFP, AFP-L3%, and DCP, the diagnostic performance of GALAD score for detecting HCC varies in population with different etiologies. Given the currently limited data, slightly lower pooled sensitivities and AUC values of GALAD score were observed in HBV populations compared to those in HCV or non-viral liver diseases populations. But previous studies also revealed a better diagnostic value of GALAD for detecting HBV-related HCC than ultrasound or other diagnostic models. The related comments have been added in the Discussion Section.

by contrast, GALAD score yielded a lowest sensitivity (0.76) and AUC (0.85) in HBV population, which means GALAD score could detect 10% more HCC patients in those with HCV or non-viral liver diseases. But compared with ultrasound or other diagnostic models constructed based on hepatitis B patients (e.g., AGED, REACH-B, PAGE-B, and mPAGE-B), GALAD still had a higher AUC value for detecting HBV-related HCC [37, 38].” Line 323-328

Point 6: the quality if the figures is very poor and must be improved.

Response 6: Thanks. We have improved the quality of the figures, and these original figures (format: TIFF) will be uploaded as supplementary materials. Thanks again.

Round 2

Reviewer 1 Report

The manuscript has not been sufficiently improved to warrant publication in JCM. The

Authors should address point by point their answers to the reviewer's comments and queries.

4 comments were outlined in the previous report.

  1. Let us begin by explaining how the true positive [TP], false positive [FP], false negative [FN], and true negative [TN] were considered in relation to which of the different gold standards used to diagnose HCC. What was considered the gold standard for diagnosing HCC in different studies: simple clinical diagnosis defined as tumor within the Barcelona Clinic liver cancer staging system or tumor within Milan criteria? Biopsy and histology? Surgical specimen? It is crucial to specify the gold standard used for every study included in your meta-analyses and to discuss any differences found among studies.

  2. Data Extraction: in the studies considered for metaanalysis I suppose that were included patients who underwent surgical procedures (liver resection, liver transplantation) or different multimodal therapies; could you specify in which proportion patients received surgical or non-surgical procedures or both?

  3. AUC: AUC represents the degree or measure of separability. It tells how well the model is able to distinguish between classes. The higher the AUC, the better the model is at predicting accuracy classes. In a Forest plot, would it be possible to visualize a second ROC discriminating for HCC in patients undergoing surgery? Surgery is usually performed for patients in an earlier stage of HCC (Milan criteria or BCLC guidelines), where accuracy is obviously more significant than accuracy in the advanced stage. This point needs to be clarified and discussed.

    4. it would be interesting to know how many false positives were found in different studies considered in your analysis.

Author Response

Point 1: Let us begin by explaining how the true positive [TP], false positive [FP], false negative [FN], and true negative [TN] were considered in relation to which of the different gold standards used to diagnose HCC. In other word specify about the pooled accuracy how was considered the gold standard for diagnosing HCC in different studies considered: simple clinical diagnosis as defined as tumor within Barcelona clinic liver cancer staging system or tumor within Milan criteria? biopsy histology? surgical specimen? It is crucial to specify the gold standard used for every study considered in your metanalysis and to discuss any difference found among studies.

Response 1: Thanks for Reviewer 1’s favorable comments. Generally, the diagnosis of a solid tumor is confirmed by histological analysis; of all solid tumors, only HCC can also be diagnosed by clinical diagnostic criteria, which is recognized at home and abroad. In all the included studies, HCC subjects were diagnosed by histological or radiological examination according to international practice guidelines (2 cohorts only by radiological examination and 1 only by histological examination), as the gold standard for evaluating the diagnostic performance of GALAD score. Without detailed data on diagnostic modality of any patient, minor differences in diagnostic gold standards among the different studies may simultaneously change the numbers of TP and FP or those of FN and TN. Similarly, in the majority of studies some CLD patients were not received further follow-up or were retested for HCC using "gold standard" tests; delayed recognition of HCC cases in the so-called control group would also mistake the two sets of values. These might lead to incorrect estimation of GALAD score performance. We have added a new column of data about diagnostic modality for HCC to Table 1 and the related comments have been described in the Discussion Section of our manuscript.

“Of all the included studies, HCC diagnosis was confirmed by histological or radiological examination according to international practice guidelines, with 2 cohorts only by radiological examination and 1 only by histological examination.” Line 145-147

“Generally, the diagnosis of a solid tumor is confirmed by histological analysis; of all solid tumors, only HCC can also be diagnosed by clinical diagnostic criteria, which is recognized at home and abroad. In all the included studies, HCC subjects were diagnosed by histological or radiological examination according to international practice guidelines, as the gold standard for evaluating the diagnostic performance of GALAD score. Without detailed data on diagnostic modality of any patient, minor differences in diagnostic gold standards among the different studies may simultaneously change the numbers of TP and FP or those of FN and TN. Similarly, in the majority of studies some CLD patients were not received further follow-up or were retested for HCC using "gold standard" tests; delayed recognition of HCC cases in the so-called control group would also mistake the two sets of values. These might lead to incorrect estimation of GALAD score performance.” Line 370-380

“Third, differences in diagnostic gold standards among the different studies may also lead to some inaccuracies in performance evaluation.” Line 419-420

Point 2: Data Extraction: in the studies considered for matanalysis I suppose that were included patients submitted to surgical procedures (liver resection, liver transplantation) or different multimodal therapies; could you specify in which proportion patients received surgical or non-surgical procedure or both?

Response 2: Thanks. For research purposes, most clinical diagnostic studies collect and analyze parameters before or at diagnosis, such as age, gender, tumor biomarkers and tumor characteristics. Without further follow-up, few parameters after diagnosis are recorded in studies, including therapeutic modalities. After carefully reviewing all the included studies, we found that only 1 study recorded the detailed proportions of HCC patients receiving different therapies, and no related data were published in the remaining 14 studies. According to current international guidelines, surgery is recommended as an optimal therapy for early-stage HCC, and non-surgical treatment for intermediated-stage or advanced-stage HCC. In other words, the distribution of tumor stages may indirectly reflect the proportions of HCC patients receiving surgical or non-surgical treatment. Usually, the distribution of tumor stages was described in most diagnostic studies so as to evaluate the performance of GALAD score for detecting early-stage HCC, and Table 2 gathered the number of patients with early-stage HCC (TP + FN) based on the definition of BCLC staging or Milan criteria. We have added a new table to Supplementary Materials and the related comments have been described in the Discussion Section of our manuscript.

“1 study recorded the detailed proportions of HCC patients receiving different therapies (Supplementary Table S2) [24].” Line 149-151

For research purposes, parameters before or at diagnosis, such as age, gender, etiologies, tumor biomarkers and tumor characteristics, are collected for analysis in most clinical diagnostic studies. Without further follow-up, few parameters after diagnosis are recorded in studies, including therapeutic modalities. Of all the included studies, only 1 study recorded the detailed proportions of HCC patients receiving different therapies. According to current international guidelines, surgery is recommended as an optimal therapy for early-stage HCC, and non-surgical treatment for intermediated-stage or advanced-stage HCC. In other words, the distribution of tumor stages, as significant parameters usually described in diagnostic studies, may indirectly reflect the proportions of HCC patients receiving surgical or non-surgical treatment.” Line 390-399

Fifth, some meaningful information (e.g., proportions of HCC patients receiving different therapies and data on GALAD for discriminating HCC in patients undergoing surgery) cannot be extracted from the majority of studies, making it difficult to evaluate its diagnostic performance from different angles.” Line 421-425

Table S2. Proportion of HCC Patients Receiving Different Therapies.

Berhane-124

Berhane-224

Berhane-324

Transplantation, %

3.6

0

4.8

Resection, %

4.2

32.9

2.6

Ablative, %

3.0

26.9

10.5

TACE, %

26.2

21.0

38.0

Sorafenib/chemotherapy, %

14.3

1.1

16.0

Supportive, %

14.3

13.5

22.7

Other palliative, %

34.5

4.6

5.4

Point 3: AUC: AUC represents the degree or measure of separability. It tells how much the model is capable of distinguishing between classes. Higher the AUC, the better the model is at predicting accuracy classes. In a Forest plot, would it be possible to visualize a second ROC discriminating for HCC in patients undergoing surgery? Surgery is usually performed for patients in an earlier stage of HCC (Milan criteria or BCLC guidelines), where accuracy is obviously more important than accuracy in the advanced stage. This point needs to be clarified and discussed.

Response 3: Thanks for the comments. For most clinical diagnostic studies, parameters before or at diagnosis, such as age, gender, etiologies, tumor biomarkers and tumor characteristics, are collected for analysis. However, without further follow-up, few parameters after diagnosis (including therapeutic modalities) are available, making it difficult to evaluate its diagnostic performance for discriminating HCC in patients undergoing surgery. According to current international guidelines, surgery is recommended as an optimal therapy for early-stage HCC (within BCLC 0/A or Milan criteria). In other words, the performance evaluation of GALAD score for detecting HCC within BCLC 0/A staging or Milan criteria could indirectly reflect its ability for discriminating HCC in patients undergoing surgery. And in Result Section, we focused on assessing the diagnostic performance of GALAD score for detecting early-stage HCC, which is more valuable than its ability for advanced-stage tumor. Our findings demonstrated a good diagnostic accuracy of GALAD score for identifying BCLC 0/A HCC with a pooled sensitivity, specificity, and AUC of 0.73 (95% CI: 0.66-0.79), 0.87 (95% CI: 0.81-0.91), and 0.86 (95% CI: 0.82-0.88), respectively. And based on the definition of Milan criteria, the pooled sensitivity, pooled specificity, and estimated AUC were 0.65 (95% CI: 0.56-0.72), 0.91 (95% CI: 0.87-0.94), and 0.87 (95% CI: 0.83-0.89), respectively. These means that a majority of HCC patients can be timely identified at their early stages by GALAD score. We have added the related comments to the Discussion Section.

For research purposes, parameters before or at diagnosis, such as age, gender, etiologies, tumor biomarkers and tumor characteristics, are collected for analysis in most clinical diagnostic studies. Without further follow-up, few parameters after diagnosis are recorded in studies, including therapeutic modalities. Of all the included studies, only 1 study recorded the detailed proportions of HCC patients receiving different therapies. According to current international guidelines, surgery is recommended as an optimal therapy for early-stage HCC, and non-surgical treatment for intermediated-stage or advanced-stage HCC. In other words, the distribution of tumor stages, as significant parameters usually described in diagnostic studies, may indirectly reflect the proportions of HCC patients receiving surgical or non-surgical treatment; and the performance evaluation of GALAD score for detecting HCC within BCLC 0/A staging or Milan criteria could indirectly indicate its ability for discriminating HCC in patients undergoing surgery. Undoubtedly, the diagnostic performance of GALAD score for detecting early-stage HCC is more valuable to be assessed compared with its ability for advanced-stage tumor. Our subgroup analyses demonstrated a good diagnostic accuracy of GALAD score for identifying BCLC 0/A HCC, with a pooled sensitivity, specificity, and AUC of 0.73 (95% CI: 0.66 - 0.79), 0.87 (95% CI: 0.81 - 0.91), and 0.86 (95% CI: 0.82 - 0.88), respectively; and based on the definition of Milan criteria, the pooled sensitivity, pooled specificity, and estimated AUC were 0.65 (95% CI: 0.56 - 0.72), 0.91 (95% CI: 0.87 - 0.94), and 0.87 (95% CI: 0.83 - 0.89), respectively. These means that a majority of HCC patients can be timely identified at their early stages by GALAD score.” Line 390-410

Fifth, some meaningful information (e.g., proportions of HCC patients receiving different therapies and data on GALAD for discriminating HCC in patients undergoing surgery) cannot be extracted from the majority of studies, making it difficult to evaluate its diagnostic performance from different angles.” Line 421-425

Point 4: It would be interesting to know how many false positives were found in different studies considered in your analysis.

Response 4: Thanks. Most study cohorts (15/19) regarded -0.63 as the cut-off value for diagnosing HCC. Among these cohorts, there were 2309 FP cases versus 13421 control patients, with FP rates ranging from 4.2% to 27.2%. As a continuous variable, the different cut-off values of GALAD score can directly change the number of FP cases; that is to say, in a fixed cohort, a FP rate decreases with the increase of cut-off value. And no uniform cut-off value setting may cause a limitation in generalizing the GALAD score to a certain degree. We have added the related description to Result Section and comments to Discussion Section. Thanks again.

Most study cohorts (15/19) regarded -0.63 as the cut-off value for diagnosing HCC. Among these cohorts, there were 2309 FP cases versus 13421 control patients, with FP rates ranging from 4.2% to 27.2% (Table 2).” Line 154-156

As a continuous variable, the different cut-off values of a diagnostic model can directly change the numbers of TP, FP, FN, and TN cases; that is to say, in a fixed study cohort, a sensitivity and FP rate will decrease with the increase of cut-off value.” Line 362-365

Reviewer 2 Report

My major comments have been addressed. I would recommend to include these considerations also in the abstract of the paper.
